# Early Repair of Rib Fractures Is Associated with Superior Length of Stay and Total Hospital Cost: A Propensity Matched Analysis of the National Inpatient Sample

**DOI:** 10.3390/medicina60010153

**Published:** 2024-01-14

**Authors:** Christopher W. Towe, Katelynn C. Bachman, Vanessa P. Ho, Fredric Pieracci, Stephanie G. Worrell, Matthew L. Moorman, Philip A. Linden, Avanti Badrinathan

**Affiliations:** 1Division of Thoracic and Esophageal Surgery, Department of Surgery, University Hospitals Cleveland Medical Center, Cleveland, OH 44106, USAavanti.badrinathan@uhhospitals.org (A.B.);; 2MetroHealth Medical Center, Department of Surgery, Division of Trauma, Critical Care, Burns, & Acute Care Surgery, Cleveland, OH 44109, USA; 3Department of Surgery Denver Health Medical Center, University of Colorado School of Medicine, Denver, CO 80045, USA; 4Division of Trauma, Critical Care and Acute Care Surgery, Department of Surgery, University Hospitals Cleveland Medical Center, Cleveland, OH 44106, USA

**Keywords:** rib fracture, rib fixation, rib plating

## Abstract

*Background and Objectives:* Previous studies have suggested that early scheduling of the surgical stabilization of rib fractures (SSRF) is associated with superior outcomes. It is unclear if these data are reproducible at other institutions. We hypothesized that early SSRF would be associated with decreased morbidity, length of stay, and total charges. *Materials and Methods:* Adult patients who underwent SSRF for multiple rib fractures or flail chest were identified in the National Inpatient Sample (NIS) by ICD-10 code from the fourth quarter of 2015 to 2016. Patients were excluded for traumatic brain injury and missing study variables. Procedures occurring after hospital day 10 were excluded to remove possible confounding. Early fixation was defined as procedures which occurred on hospital day 0 or 1, and late fixation was defined as procedures which occurred on hospital days 2 through 10. The primary outcome was a composite outcome of death, pneumonia, tracheostomy, or discharge to a short-term hospital, as determined by NIS coding. Secondary outcomes were length of hospitalization (LOS) and total cost. Chi-square and Wilcoxon rank-sum testing were performed to determine differences in outcomes between the groups. One-to-one propensity matching was performed using covariates known to affect the outcome of rib fractures. Stuart–Maxwell marginal homogeneity and Wilcoxon signed rank matched pair testing was performed on the propensity-matched cohort. *Results:* Of the 474 patients who met the inclusion criteria, 148 (31.2%) received early repair and 326 (68.8%) received late repair. In unmatched analysis, the composite adverse outcome was lower among early fixation (16.2% vs. 40.2%, *p* < 0.001), total hospital cost was less (USD114k vs. USD215k, *p* < 0.001), and length of stay was shorter (6 days vs. 12 days) among early SSRF patients. Propensity matching identified 131 matched pairs of early and late SSRF. Composite adverse outcomes were less common among early SSRF (18.3% vs. 32.8%, *p* = 0.011). The LOS was shorter among early SSRF (6 days vs. 10 days, *p* < 0.001), and total hospital cost was also lower among early SSRF patients (USD118k vs. USD183k late, *p* = 0.001). *Conclusion:* In a large administrative database, early SSRF was associated with reduced adverse outcomes, as well as improved hospital length of stay and total cost. These data corroborate other research and suggest that early SSRF is preferred. Studies of outcomes after SSRF should stratify analyses by timing of procedure.

## 1. Background

The optimal care of patients with rib fractures has significantly evolved in the last two decades. In the landmark trial by Tanaka et al., the surgical stabilization of rib fractures (SSRF) was only performed after a period of five days of mechanical ventilation [1]. Since then, guidelines and randomized trials have suggested that SSRF should be carried out within 72 h of injury [2,3]. However, recent studies on the optimal timing of SSRF indicated that even earlier intervention was associated with improved outcomes [4]. In one study, patients with multiple rib fractures or a flail chest were retrospectively analyzed based on the timing of SSRF. The analysis revealed that patients receiving early SSRF after the injury were less than half as likely to develop pneumonia, a third as likely to require prolonged mechanical ventilation, and showed a trend toward lower rates of tracheostomy [5,6].

Unfortunately, these studies have faced criticism for methodological flaws and other issues, which raise questions about the generalizability of the findings to other medical centers [5]. The variation in institutional practices regarding the timing of surgical stabilization of rib fractures introduces an additional layer of complexity to the ongoing discourse. For example, all SSRF procedures were performed at high-volume trauma centers, which may not accurately reflect the care provided at other institutions. The rate of “early” fixation also increased over the study period, which raises questions about whether other factors during the study period confounded these findings of the benefit of early intervention. Factors such as institutional resources, expertise, and patient demographics may contribute to this variability. Therefore, the question of the optimal timing of SSRF remains unclear. On one hand, early intervention may be associated with shorter overall hospitalization, whereas later intervention may allow for better preoperative “optimization” before surgery.

A comprehensive review of the existing literature reveals a paucity of studies directly comparing the outcomes of early and late SSRF, particularly in a larger nationally representative patient cohort. While numerous investigations have explored the overall benefits of surgical intervention in rib fractures, the specific impact of the timing of SSRF remains a relatively understudied topic. The purpose of this study is to compare surgical outcomes of patients receiving early vs. late SSRF, using statistical methods to correct for differences between the groups. We hypothesized that early rib fixation would be associated with improved morbidity, hospital length of stay, and total hospital cost in patients presenting with rib fractures.

## 2. Methods

### 2.1. Data Source

This is a retrospective database analysis of the National Inpatient Sample (NIS) and Healthcare Cost and Utilization Project (HCUP). The NIS and HCUP, compiled by the Agency for Healthcare Research and Quality, were used to analyze patients who received SSRF in the fourth quarter of 2015 through 2016 [7]. The NIS is a weighted sample of hospital admissions across the United States and includes data from inpatient stays, not individual patients. It captures conditions, procedures, and diagnoses occurring during a specific inpatient hospital encounter. No individual patients are identifiable through this database. Records of events and diagnoses before or after the stay are not available and are not included in this analysis.

### 2.2. Study Population

Adult patients (>18 years old) were included in the analysis if they had a diagnosis of multiple rib fractures or a flail chest and received SSRF. The diagnosis of multiple fractures or a flail chest was based on the ICD-10 codes, which included (for multiple rib fractures) S22.4, S22.41, S22.42, S22.43, S22.49, and (for a flail chest) S22.5. Patients with diagnoses of multiple rib fractures and a flail chest were categorized as having a flail. Receipt of SSRF was defined by ICD-10-PCS codes, including code “bases”: OPB1, OPB2, OPH1, OPH2, OPP1, OPP2, OPQ1, OPQ2, OPR1, OPR2, OPS2, OPT1, OPT2, OPU1, OPU2, OPW1, and OPW2. Patients were excluded for missing critical study variables, including procedure day, length of stay, or total hospital charge. Patients were also excluded for concurrent receipt of neurosurgery (as per ICD-10-PCS code) or a diagnosis of traumatic brain injury (as per HCUP clinical classification software combined for ICD-10-CM diagnosis) during their hospitalization.

Patients were categorized as having early or late SSRF based on the timing of their surgical procedure. Early was defined as SSRF procedures occurring on or before hospital day 1, and late SSRF was defined as SSRF procedures performed on or after hospital day 2 and before hospital day 11.

Demographic variables, comorbidities, and injury severity score (ISS) were also extracted from the NIS database. Age is uniformly truncated at age 90, with all patients greater than 90 years old analyzed as 90. Comorbidity was characterized by the Elixhauser measure, which is a categorization of comorbidities similar to the Charlson index and is a frequently used method to identify comorbid conditions in administrative databases [8,9]. The Elixhauser variables were extracted based on ICD-10 diagnosis coding using HCUP comorbidity software (HCUP Comorbidity Software, 2008). ISS score was also estimated using ICD-10 codes using the open-access program ICDPIC-R [10,11]. ISS was further categorized into mild (≤8), moderate (9–15), severe (16–25), and profound (≥25) [12]. The mechanism of injury was categorized as motor vehicle traffic (EXT007) or other using external cause of injury codes.

### 2.3. Outcome Variables

The primary outcome of interest was a composite of adverse outcomes previously demonstrated to be associated with rib fractures: death, pneumonia, tracheostomy, or discharge to a short-term acute care hospital. The NIS provides information on whether in-hospital death occurred. Pneumonia and tracheostomy were collected from ICD-10 codes and only represent complications during the primary admission. Secondary outcomes were length of hospitalization (LOS) and total hospital charge. Readmission data were not collected in the NIS and were therefore not analyzed.

### 2.4. Statistical Analysis

Demographic and clinical characteristics were summarized using descriptive statistics and compared between early and late SSRF groups using univariate analyses (chi-square test or rank-sum test as appropriate). Effect sizes were calculated using Cohen’s d testing. A multivariable logistic regression was performed to evaluate the association of early SSRF with the adverse composite outcome, using additional covariates determined through backward selection from univariate modeling with an α ≤ 0.05. An additional least squares linear regression was performed, using the same covariates, to evaluate the secondary outcomes of LOS and total hospital cost.

To further reduce possible confounding effects of these variables on the outcomes of interest, we performed propensity score matching using covariates that differed or were clinically significant between the early and late SSRF groups. Variables included in the propensity match were age (≥65 vs. <65), diabetes, weight loss, coagulopathy, electrolyte abnormality, flail chest, mechanism of injury motor vehicle traffic, and ISS category. Nearest neighbor 1 to 1 matching with no replacement was used, with the caliper set to 0.05. Differences in the presence of the composite adverse outcome were compared using the Stuart–Maxwell test of homogeneity, and differences in the length of stay and total hospital charge were compared using the Wilcoxon matched-pairs signed rank test.

Finally, we evaluated the outcomes of SSRF at the institutional level. Institutions with more than four SSRF procedures in the database were included. An “early SSRF ratio” was calculated, which was the number of early SSRF procedures divided by the total SSRF procedures. Institutions were considered “high-use early SSRF institutions” if the early SSRF ratio was ≥0.5.

All statistical analyses were performed using Stata version 16.0 statistical software (Statacorp, College Station, TX, USA). This study was considered exempt from IRB approval because the data are de-identified.

## 3. Results

A total of 474 patients met the inclusion criteria and underwent SSRF for multiple fractures or a flail chest, representing 2370 inpatient hospitalizations. Among the 474 patients, 148 (31.2%) received early repair, and 326 (68.8%) received late repair. The distribution of procedures by hospital day is shown in Figure 1. The characteristics of the cohort are presented in Table 1, stratified by SSRF timing. The demographic and injury characteristics varied based on the timing of SSRF. Notably, patients receiving early SSRF were less likely to have a flail chest (early 45/148 (30.4%) vs. late 175/326 (53.7%), *p* < 0.001), and they were more likely to have a mild ISS category (early 68/148 (46.0%) vs. late 63/326 (19.3%), *p* < 0.001). Flail chest also demonstrated the largest effect size in unmatched analyses (Table 1).

Among the 474 patients who underwent SSRF in this cohort, the average hospital length of stay was 13 days, and 8 patients died (1.7%). The primary composite adverse outcome occurred in 155 patients (32.7%), and it was less common among patients who received early repair (early 24/148 (16.2%) vs. late 131/326 (40.2%), *p* < 0.001). Other demographic and injury characteristics also influenced the occurrence of adverse outcomes (Table 2).

### 3.1. Multivariable Analysis

To reduce potential confounding, we performed a multivariable logistic regression (Table 3), which demonstrated an independent effect of early SSRF timing in reducing the composite adverse outcome (OR 0.42, 95% CI 0.20–0.87, *p* = 0.020). Hospital length of stay was shorter in the early SSRF group in unadjusted analysis (early 6 days vs. late 12 days, *p* < 0.001) and in a multivariable regression model (coefficient −4.66, 95% CI −6.54 to −2.79, *p* < 0.001) (Table 4). Total hospital cost was also lower after early SSRF, in an unadjusted analysis, (early USD113,826 vs. late USD215,180, *p* < 0.001), and there was a trend towards a reduced cost in a multivariable regression model (coefficient −45,162, 95% CI −91,459 to −1134, *p* = 0.056) (Table 4).

### 3.2. Propensity Matched Analysis

Propensity matching identified 131 matched pairs of early and late SSRF. Propensity matching effectively reduced bias between the groups (median bias unmatched 28.4 vs. matched 2.1) (Figure 2). The composite adverse outcome was less common among patients receiving early SSRF (early 24/131 (18.3%) vs. late 43/131 (32.8%), *p* = 0.011). LOS was shorter among early SSRF (early 6 days (IQR 4–10) vs. late 10 days (IQR 8–16), *p* < 0.001), and total hospital cost was also lower among early SSRF patients (early USD118,342 (IQR 73,779–209,826) vs. late USD182,659 (IQR 127,754–278,090), *p* < 0.001).

## 4. Discussion

This is a retrospective national database analysis, demonstrating that early rib fracture repair is associated with reduced morbidity, a decreased length of hospital stay, and lower total hospital costs. Several reasons suggest that prompt rib fracture repair might be advantageous. Rib fractures can lead to respiratory physiology abnormalities, and stabilizing the injury quickly may reduce the “burden” of these abnormalities on patients, thus reducing respiratory complications associated with fractures. A previous study has shown that early SSRF was associated with a shorter operating room duration [4], suggesting that patients receiving early procedures may benefit from a reduced duration of anesthesia associated with the SSRF procedure and/or reduced procedural complexity. Recent trials have demonstrated improved outcomes in patients with rib fracture as well, including decreased length of stay and ICU length stay [13]. As such, the findings of this study further reaffirm our understanding of the benefits of early rib fixation.

From a healthcare system perspective, early repair is also likely to be economically beneficial. This study shows that early SSRF is associated with a shorter hospital length of stay and reduced total hospital costs. One important aspect that merits consideration is the potential compounding effect of early rib fracture repair on the broader healthcare system. By reducing hospital length of stay and associated costs, early intervention not only benefits individual patients but also contributes to resource allocation and optimization. Hospitals can potentially allocate resources more efficiently, leading to increased capacity to address other pressing medical needs. This is especially pertinent in the context of healthcare systems facing challenges such as overcrowding and resource constraints. The economic advantages of early rib fixation, as demonstrated in our study, may thus extend beyond the immediate benefits to patients, presenting an opportunity for healthcare institutions to enhance overall system efficiency and resilience. Similarly, shorter hospitalization is associated with reduced patient morbidities, as it can reduce the risk of nosocomial infection and iatrogenesis. These findings corroborate another retrospective study on the optimal timing of SSRF in multiple rib fractures and flail chest, which suggested that early SSRF was associated with a shorter length of hospitalization and lower rates of pneumonia and prolonged mechanical ventilation [5]. Another similar retrospective trial, which defined early fixation as less than 48 h, also demonstrated that early SSRF was associated with improvements in hospital length of stay, reduced rates of pneumonia, and fewer tracheostomies [14]. Our study addresses the methodological concerns that existed with previous studies and demonstrates that their previous findings are reproducible in larger nationally representative sample.

The current recommendation for patients with flail chest after blunt trauma who are appropriate surgical candidates is SSRF within 72 h after injury [4]. This recommendation is based on studies comparing the outcomes of patients offered SSRF within 24–72 h after injury versus those offered SSRF 72 h after injury. In our study, early SSRF was defined as a procedure performed on hospital day 0 or 1, which is within the time recommended. These findings, in combination with other retrospective reports, suggest that SSRF should be performed earlier than current guidelines. We believe that the benefit of early fixation may operate as a “continuous” variable, with even shorter times associated with some benefit. For example, sepsis data demonstrate the benefit of rapid administration of antibiotics, and similar factors may also be present in patients with rib fractures [15]. Certainly, there are practical concerns with early SSRF that warrant consideration. Patients should be stabilized prior to intervention, and clinical judgment should determine the optimal timing for a given patient and injury. In turn, the optimal timing of SSRF necessitates a collaborative effort involving various medical disciplines, including trauma surgeons, anesthesiologists, respiratory therapists, and intensive care specialists. The coordination of care among these professionals is integral to achieving optimal outcomes.

Furthermore, the consideration of patient-centered outcomes is crucial when evaluating the impact of early rib fracture repair. While our study extensively addresses reductions in morbidity, hospital length of stay, and overall costs, it is essential to explore the qualitative aspects of patient recovery. Future research could delve into assessing patient-reported outcomes, pain management, and functional recovery after early rib fixation. Understanding the patient experience and satisfaction with early intervention will provide a more comprehensive picture of the benefits and potential trade-offs associated with this approach. By incorporating patient perspectives into the decision-making process, we can better tailor interventions to meet the holistic needs of individuals recovering from rib fractures and enhance the overall quality of trauma care.

As we advocate for the potential advantages of early rib fracture repair, it is crucial to acknowledge the evolving landscape of surgical techniques and technologies. Advances in surgical approaches, instrumentation, and postoperative care may further enhance the feasibility and safety of early interventions. Ongoing research and technological innovations could potentially address some of the practical concerns associated with early rib fixation, providing clinicians with additional tools to optimize patient outcomes. Collaborative efforts between clinicians, researchers, and industry stakeholders are essential in driving forward these advancements and refining the optimal strategies for managing rib fractures in the evolving landscape of trauma care. As we strive for continuous improvement in trauma care, considering the systemic impact of early interventions and remaining attuned to advancements in surgical approaches will be crucial in shaping the future landscape of rib fracture management.

### Limitations

We recognize that this study has several limitations. Firstly, as a large retrospective database study, we are unable to verify the accuracy of individual entries within the databases utilized. Additionally, we cannot control for unmeasured variables that are not included in the NIS. For example, certain injury characteristics are not recorded and may bias the analysis. Patients who received early surgery may have been less severely injured than those in the “late” group and may have had a more successful recovery regardless of the timing of their surgery. We have attempted to control for this by including ISS in the propensity match. Furthermore, we cannot control for other variables that may have appropriately delayed surgical intervention, such as ongoing resuscitation from other injuries. We have attempted to correct for this bias by analyzing outcomes between programs, though the impact of concurrent injuries may exit. In addition, we cannot evaluate complications or deaths that occur after discharge, and this may have altered or biased our findings. Lastly, we are unable to assess the specific number of rib fractures that underwent SSRF, as data derived from billing and coding information do not denote the specific number of fractures. Despite these limitations, we believe that this study highlights the improved patient outcomes and cost benefit of early rib fixation in patients presenting with rib fractures.

## 5. Conclusions

In conclusion, this large retrospective study of a nationally representative sample demonstrates the benefit of early rib fracture repair in patients presenting with rib fractures. Early rib fracture repair, within a day of presentation, is associated with reduced morbidity, shorter hospital stays and lower overall costs. As such, we advocate for a proactive approach that prioritizes timely rib fixation in patients with rib fractures when clinically appropriate for improved patient outcomes and healthcare costs.

## Figures and Tables

**Figure 1 medicina-60-00153-f001:**
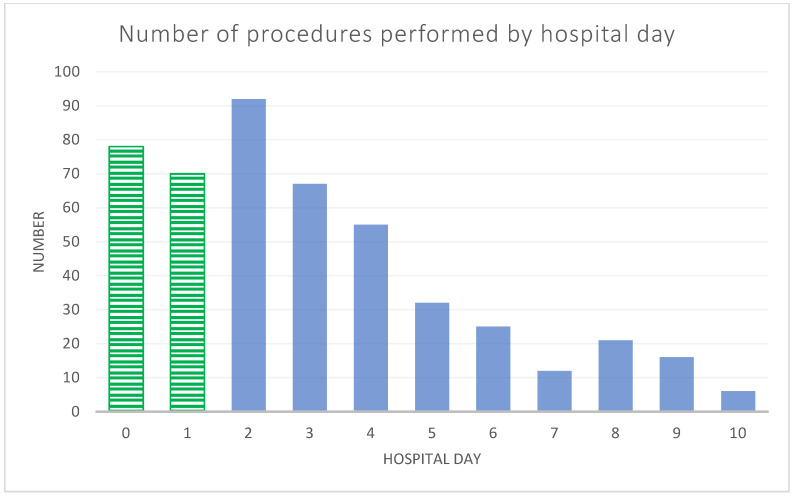
Number of rib fixation procedures performed per hospital day among 447 patients with multiple rib fractures or flail chest in the National Inpatient Sample (NIS) in 2016. Procedures performed on hospital days 0–1 were categorized as “early” (green stripe) and those on days 2–10 were categorized as “late” (solid blue).

**Figure 2 medicina-60-00153-f002:**
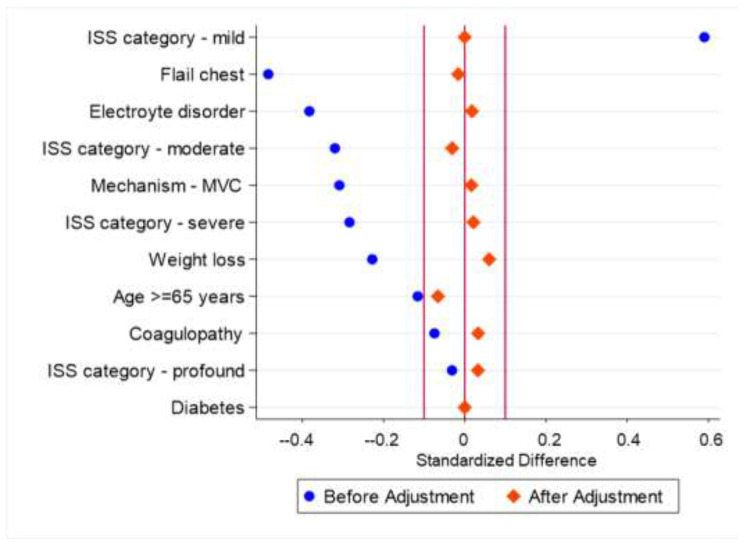
Effect of propensity matching on standardized difference between cohorts receiving early (hospital day 0–1) vs. late (hospital day 2–10) surgical rib stabilization.

**Table 1 medicina-60-00153-t001:** Relationship of demographic variables to timing of surgical fixation of rib fractures in the National Inpatient Sample (NIS). Patients categorized as “early” if procedures were performed on hospital days 0–1 or “late” if performed on days 2–10. Comparisons between groups were performed using chi square for categorical variables or rank-sum test for continuous variables.

Category		Early SSRF N = 148	Late SSRF N = 326	*p*-Value	Effect Size (Estimate, 95% Confidence Interval)
Age (years)		55.5 (45–66)	57 (45–69)	0.321	0.094 (−0.100–0.288)
Female sex		44 (29.7%)	80 (29.7%)	0.234	−0.118 (−0.312–0.76)
Academic institution		121 (81.8%)	285 (87.4%)	0.103	0.161 (−0.032–0.356)
Private insurance		64 (43.2%)	123 (37.9%)	0.266	−0.110 (−0.304–0.084)
Hypertension		66 (44.6%)	144 (44.2%)	0.932	−0.008 (−0.202–0.185)
Diabetes		20 (13.5%)	44 (13.5%)	0.996	−0.040 (−0.232–0.153)
Chronic lung disease		22 (14.9%)	44 (13.5%)	0.690	−0.039 (−0.233–0.154)
Peripheral vascular disease		12 (3.7%)	7 (4.7%)	0.590	−0.053 (−0.247–0.140)
Congestive heart failure		20 (6.1%)	6 (4.1%)	0.356	0.091 (−0.103–0.285)
Weight loss		32 (9.8%)	6 (4.1%)	0.032	0.212 (0.017–0.407)
Coagulopathy		21 (6.4%)	6 (4.1%)	0.464	0.072 (−0.121–0.266)
Electrolyte disorder		26 (17.6%)	111 (34.1%)	<0.001	0.368 (0.172–0.563)
Flail chest		45 (30.4%)	175 (53.7%)	<0.001	0.477 (0.280–0.673)
Mechanism: MVC		39 (26.4%)	133 (40.8%)	0.002	0.302 (0.107–0.497)
Injury severity score		9 (1–11)	10 (9–17)	<0.001	0.471 (0.275–0.668)
ISS category	Mild	68 (46.0%)	63 (19.3%)	<0.001	−0.671 (−0.815–0.419)
	Moderate	57 (38.5%)	177 (54.3%)	0.001	0.318 (0.122–0.513)
	Severe	15 (10.1%)	66 (20.3%)	0.007	0.270 (0.074–0.465)
	Profound	8 (5.4%)	20 (6.1%)	0.755	0.030 (−0.163–0.225)

Abbreviations: SSRF: surgical stabilization of rib fractures, MVC: motor vehicle traffic, ISS: injury severity score.

**Table 2 medicina-60-00153-t002:** Relationship of demographic variables to adverse outcomes in the National Inpatient Sample (NIS). Patients were categorized as having an adverse outcome for the occurrence of a composite outcome of death, pneumonia, tracheostomy, or discharge to a short-term acute care hospital. Comparisons between groups were performed using chi square for categorical variables or rank-sum test for continuous variables.

Category		Adverse Outcome If Factor Present	Adverse Outcome If Factor Absent	*p*-Value
Early SSRF (vs. late)		24/148 (16.2%)	131/326 (40.2%)	<0.001
Age > 65 years		57/153 (37.3%)	110/341 (32.3)	0.278
Female sex		36/126 (28.6%)	131/368 (35.6%)	0.150
Academic institution		143/422 (33.9%)	24/72 (33.3%)	0.927
Private insurance		67/196 (34.2%)	100/297 (33.7%)	0.906
Hypertension		65/213 (30.5%)	102/281 (36.3%)	0.178
Diabetes		15/65 (23.1%)	152/429 (35.4%)	0.050
Chronic lung disease		23/67 (34.3%)	144/427 (33.7%)	0.923
Peripheral vascular disease		6/19 (31.6%)	161/475 (33.9%)	0.834
Congestive heart failure		12/27(44.4%)	155/467 (33.2%)	0.229
Weight loss		28/42 (66.7%)	139/452 (30.8%)	<0.001
Coagulopathy		17/30 (56.7%)	150/464 (32.3%)	0.006
Electrolyte disorder		72/149 (48.3%)	95/345 (27.5%)	<0.001
Flail chest		105/234 (44.9%)	62/260 (23.9%)	<0.001
Mechanism: MVC		79/181 (43.7%)	88/313 (28.1%)	<0.001
ISS category	Mild	17/132 (12.9%)	150/362 (41.4%)	<0.001
	Moderate	95/243 (39.1%)	72/251 (28.7%)	0.014
	Severe	39/90 (43.3%)	128/404 (31.7%)	0.035
	Profound	16/29 (55.2%)	151/465 (32.5%)	0.012

Abbreviations: SSRF: surgical stabilization of rib fractures, MVC: motor vehicle traffic, ISS: injury severity score.

**Table 3 medicina-60-00153-t003:** Multivariable logistic regression of relationship of early surgical stabilization of rib fractures to adverse outcome, defined as the composite of death, pneumonia, tracheostomy, or discharge to a short-term acute care hospital, in the National Inpatient Sample (NIS).

Covariate		Odds Ratio (OR)	95% Conf. Interval	*p*-Value
Early SSRF (vs. late)		0.43	0.21–0.87	0.020
Diabetes		0.72	0.3–1.72	0.458
Weight loss		3.65	1.37–9.71	0.010
Coagulopathy		2.24	0.81–6.25	0.121
Electrolyte disorder		1.71	0.93–3.14	0.083
Flail chest (vs. multiple rib fracture)		1.48	0.88–2.48	0.136
Mechanism: MVC (vs. other)		1.46	0.83–2.56	0.191
ISS category	Mild	ref	-	-
	Moderate	2.37	1.13–4.99	0.023
	Severe	2.65	1.02–6.87	0.045
	Profound	4.85	1.33–17.66	0.017

Abbreviations: SSRF: surgical stabilization of rib fractures, MVC: motor vehicle traffic, ISS: injury severity score.

**Table 4 medicina-60-00153-t004:** Multivariable linear regression of relationship of early surgical stabilization of rib fractures to hospital length of stay and total hospital charge, in the National Inpatient Sample (NIS).

Covariate		Hospital Length of Stay			Total Hospital Charge		
		Coefficient	95% Conf. Interval	*p*-Value	Coefficient	95% Conf. Interval	*p*-Value
Early SSRF (vs. late)		−4.67	−6.54–−2.79	0.000	−45,162	−91,460–1135	0.060
Diabetes		−1.85	−3.73–0.03	0.050	−59,860	−101,318–−18,401	0.010
Weight loss		7.22	2.93–11.52	0.000	168,451	14,942–321,960	0.030
Coagulopathy		2.83	−0.68–6.35	0.110	52,971	−40,157–146,098	0.260
Electrolyte disorder		2.46	0.36–4.57	0.020	43,819	−10,153–97,792	0.110
Flail chest (vs. multiple rib fracture)		1.52	−0.41–3.46	0.120	31,321	−15,304–77,946	0.190
Mechanism: MVC (vs. other)		2.66	0.71–4.61	0.010	87,187	36,446–137,929	0.000
ISS category	Mild	ref	-	-	ref	-	-
	Moderate	0.69	−1.54–2.92	0.540	37,641	−14,083–89,364	0.150
	Severe	2.58	−0.44–5.6	0.090	75,453	4686–146,219	0.040
	Profound	3.76	−0.2–7.72	0.060	70,398	−42,302–183,097	0.220

## Data Availability

Upon request, the corresponding author can provide the data utilized in this study.

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
