# Peer review of "Early Repair of Rib Fractures Is Associated with Superior Length of Stay and Total Hospital Cost: A Propensity Matched Analysis of the National Inpatient Sample"

_medicina, 2024, doi:10.3390/medicina60010153_

Round 1

Reviewer 1 Report

Comments and Suggestions for Authors

Dear Authors,

Thank you for the effort you put into your research. I have read and examined your research in detail. Although the research is a good topic when evaluated in terms of subject and scope, there are big problems in the writing phase. For now, I have to decline your inquiry. I think you can upload your research again after the edits I will give below.

Best.

First of all, the entire article is written in different formats and different writing styles, making it difficult to understand while reading. It must fully comply with the journal rules.

Introduction

First of all, the introduction is very shallow and devoid of literature. For this reason, I recommend that you go into this section more specifically and emphasize more clearly what similar research has revealed in this field and what your research will contribute to the literature beyond these studies. Briefly, present this section with more sources by going from general to specific to your main hypothesis.

method

In this section, I recommend that you make the chapter more understandable to your readers by dividing it into categories and also mentioning and visualizing surgical procedures.

Results

In addition to the p values for your paired comparison groups, highlight the effect levels by using formulas such as cohend effect size to talk about the effects of the study, even if they are not significant. I don't understand why you used the chi square test in pairwise comparisons. Wouldn't pairwise group comparison tests make more sense?

I also recommend that you present your graphics in a more understandable way by evaluating them in higher quality graphics programs.

Why did you make a conclusion section directly after the findings? If this section is a discussion, did you do it using only two references (4 and 11)? Please conclude with a good discussion of your results, supported by the literature, followed by limitations and conclusion.

In short, in my opinion, your research needs to be revised from beginning to end.

Best.

Author Response

Thank you for the effort you put into your research. I have read and examined your research in detail. Although the research is a good topic when evaluated in terms of subject and scope, there are big problems in the writing phase. For now, I have to decline your inquiry. I think you can upload your research again after the edits I will give below.

 Best.

 First of all, the entire article is written in different formats and different writing styles, making it difficult to understand while reading. It must fully comply with the journal rules. 

We have completely edited and rewritten the manuscript for improved readability.

 Introduction

 First of all, the introduction is very shallow and devoid of literature. For this reason, I recommend that you go into this section more specifically and emphasize more clearly what similar research has revealed in this field and what your research will contribute to the literature beyond these studies. Briefly, present this section with more sources by going from general to specific to your main hypothesis. 

Thank you. Further literature has been cited

 method

 In this section, I recommend that you make the chapter more understandable to your readers by dividing it into categories and also mentioning and visualizing surgical procedures. 

Thank you. Categories have been added to aid in readability

 Results

 In addition to the p values for your paired comparison groups, highlight the effect levels by using formulas such as cohend effect size to talk about the effects of the study, even if they are not significant. I don't understand why you used the chi square test in pairwise comparisons. Wouldn't pairwise group comparison tests make more sense? 

Chi square tests were used for categorical variables in pairwise comparisons. In addition, a Cohen’s D test was performed for all variables with the values displayed in table 1 and significant values discussed in results.

 I also recommend that you present your graphics in a more understandable way by evaluating them in higher quality graphics programs. 

Thank you, we have included a higher quality image

 Why did you make a conclusion section directly after the findings? If this section is a discussion, did you do it using only two references (4 and 11)? Please conclude with a good discussion of your results, supported by the literature, followed by limitations and conclusion. 

 In short, in my opinion, your research needs to be revised from beginning to end

Thank you for your insightful comments. We have taken them to heart and hope our revisions are satisfactory.

Reviewer 2 Report

Comments and Suggestions for Authors

Thank you for the opportunity to review this manuscript “Early repair of rib fractures is associated with superior length of stay and total hospital cost: a propensity matched analysis of the National Inpatient Sample.”

I have a couple questions dash comments.

My main concern is the composite outcome of death, pneumonia, and discharge to short term hospital. I would like to ask the authors to better explain why this composite outcome was selected. It would be very helpful if the authors can provide references for the previous studies that you used with composite outcome. Can you please report separately results for death, pneumonia, and discharge to short term hospital? If you cannot do it please explain the reason why not.

I did not find Discussion but only Conclusion sections. Based on my knowledge of this journal, having the discussion section is a requirement.

As one of limitations of the study I would also state the number of rib fractures that underwent SSRF is not known and cannot be accounted for in the analyses.

How many patients were excluded due to missing data? Please report it in the results section.

Please elaborate on why you decided to choose this time cut off for early versus late surgical refixation? The figure #1 demonstrates that majority of the patients underwent SSRF was the first four days. Did you do subgroup analysis of patients who underwent SSRF on day 2-4? if not, can you please do this and report. if no differences between early and day 2-4 will be found it may change the recommendation in terms of timing of surgical refixation.

I would also ask to expand more on the definitions of early versus late SSRF is it has been reported in the previous surgical literature. I am not sure there is a big difference between day 1 and 2.

Please make sure report OR in the same manner in the text and tables, I am referring to an accuracy of decimal points.

Thank you

Author Response

Thank you for the opportunity to review this manuscript “Early repair of rib fractures is associated with superior length of stay and total hospital cost: a propensity matched analysis of the National Inpatient Sample.”

I have a couple questions dash comments. 

My main concern is the composite outcome of death, pneumonia, and discharge to short term hospital. I would like to ask the authors to better explain why this composite outcome was selected. It would be very helpful if the authors can provide references for the previous studies that you used with composite outcome. Can you please report separately results for death, pneumonia, and discharge to short term hospital? If you cannot do it please explain the reason why not.

Unfortunately, this outcome was used as a composite given the few patients who actually sustained this outcome.

I did not find Discussion but only Conclusion sections. Based on my knowledge of this journal, having the discussion section is a requirement. We have reformatted to adhere to this requirement

As one of limitations of the study I would also state the number of rib fractures that underwent SSRF is not known and cannot be accounted for in the analyses. We have addressed this in limitations

How many patients were excluded due to missing data? Please report it in the results section. 

There were no further rib fracture patients excluded from the dataset. Patient were excluded for not meeting the criteria of the cohort for this study.

Please elaborate on why you decided to choose this time cut off for early versus late surgical refixation? The figure #1 demonstrates that majority of the patients underwent SSRF was the first four days. Did you do subgroup analysis of patients who underwent SSRF on day 2-4? if not, can you please do this and report. if no differences between early and day 2-4 will be found it may change the recommendation in terms of timing of surgical refixation. 

While we could perform this analysis, the recommended time by which to perform SSRF is 72 hours so the day 3-4 patients. We addressed the discrepancies between groups in our propensity match

I would also ask to expand more on the definitions of early versus late SSRF is it has been reported in the previous surgical literature. I am not sure there is a big difference between day 1 and 2.

Please make sure report OR in the same manner in the text and tables, I am referring to an accuracy of decimal points. 

We have corrected these in the body of the text.

Reviewer 3 Report

Comments and Suggestions for Authors

Dear authors,  thank you for submitting the paper.

I just have a few small questions:

Can you give international readers more informations about the way the codes and informations are recorded is the data base. Is it possible not to code the operations or diagnoses.

Are also all transvered patients to a special unit or hospital be identified and included?

As you mentioned the bias of severe co-injuries or illnesses are not be clearly checked  and identified - the conclusion might be a bit more precised on this. 

Author Response

Can you give international readers more informations about the way the codes and informations are recorded is the data base. Is it possible not to code the operations or diagnoses.

As described in the methods section, ICD-10 coding was used to identify operations and diagnoses. This coding scheme is used primarily for billing and is unable to distinguish between specific injury patterns beyond what is coded and cannot distinguish the specifics of procedures beyond the specified categories.

Are also all transvered patients to a special unit or hospital be identified and included? Patients are identified by the facilities at which they are admitted in an aggregate fashion.

As you mentioned the bias of severe co-injuries or illnesses are not be clearly checked  and identified - the conclusion might be a bit more precised on this.

Thank you, a clarification has been added  

Reviewer 4 Report

Comments and Suggestions for Authors

I congratulate the authors for their retrospective study, collecting 474 patients undergoing surgical stabilization of rib fractures (SSRF), to investigate whether early SSRF would be associated with decreased morbidity, length of stay, and total charges.

Based on the previous study for rib fracture stabilization, the authors’ examined the superiority of early SSRF over late one, and reconfirmed that early SSRF has significant effect on reducing adverse composite outcome. Fortunately, the results of the study presented have been proven to be reproducible at any institutional level. 

I think the technical contents (Background, Methods, Results, Conclusion etc.) of the submitted manuscript are well written and summarized.

I have a few relatively minor comments, explained below;

1.      In page 10, Conclusion section, the authors describe “The current recommendation for patients with flail chest after blunt trauma who are appropriate surgical candidates is SSRF less than 72 hours after injury”. The authors should cite the paper regarding the current recommendation, as is done in Background section.

2.      Regarding the definition of early and late SSRF.

In the study presented, early was defined as SSRF procedures occurring on or before hospital day 1 and late SSRF as SSRF performed on or after hospital day 2. Could the authors describe any specific reason for that definition for early and late SSRF in the study?

I hope my comments would be helpful for the authors.

Author Response

I think the technical contents (Background, Methods, Results, Conclusion etc.) of the submitted manuscript are well written and summarized.

I have a few relatively minor comments, explained below;

  1. In page 10, Conclusion section, the authors describe “The current recommendation for patients with flail chest after blunt trauma who are appropriate surgical candidates is SSRF less than 72 hours after injury”. The authors should cite the paper regarding the current recommendation, as is done in Background section. This citation has been added

  2. Regarding the definition of early and late SSRF.

In the study presented, early was defined as SSRF procedures occurring on or before hospital day 1 and late SSRF as SSRF performed on or after hospital day 2. Could the authors describe any specific reason for that definition for early and late SSRF in the study? 

Within the appropriate window, we chose these definitions to delineate the actual impact of earlier operation while still remaining within the suggested window of 72 hours.

I hope my comments would be helpful for the authors